

# Hidden diversity: comparative functional morphology of humans and other species

Erin A. McKenney[1,2,*], Amanda R. Hale[3,7,*], Janiaya Anderson[4], Roxanne Larsen[5,6], Colleen Grant[3] and Robert R. Dunn[1]

[1] Department of Applied Ecology, North Carolina State University, Raleigh, NC, United States of America
[2] North Carolina Museum of Natural Sciences, Raleigh, NC, United States of America
[3] Department of Biological Sciences, North Carolina State University, Raleigh, NC, United States of America
[4] Department of Psychology, North Carolina State University, Raleigh, NC, United States of America
[5] Office of Curricular Affairs, Duke University School of Medicine, Durham, NC, United States of America
[6] Department of Veterinary and Biomedical Sciences, University of Minnesota, Saint Paul, MN, United States of America
[7] SNA International for the Defense POW/MIA Accounting Agency, Joint Base Pearl Harbor-Hickam, HI, United States of America
[*] These authors contributed equally to this work.

Corresponding author
Erin A. McKenney,
eamckenn@ncsu.edu

## ABSTRACT

Gastrointestinal (GI) morphology plays an important role in nutrition, health, and epidemiology; yet limited data on GI variation have been collected since 1885. Here we demonstrate that students can collect reliable data sets on gut morphology; when they do, they reveal greater morphological variation for some structures in the GI tract than has been documented in the published literature. We discuss trait variability both within and among species, and the implications of that variability for evolution and epidemiology. Our results show that morphological variation in the GI tract is associated with each organ's role in food processing. For example, the length of many structures was found to vary significantly with feeding strategy. Within species, the variability illustrated by the coefficients of variation suggests that selective constraints may vary with function. Within humans, we detected significant Pearson correlations between the volume of the liver and the length of the appendix ($t$-value = 2.5278, df = 28, $p = 0.0174$, corr = 0.4311) and colon ($t$-value = 2.0991, df = 19, $p = 0.0494$, corr = 0.4339), as well as between the lengths of the small intestine and colon ($t$-value = 2.1699, df = 17, $p = 0.0445$, corr = 0.4657), which are arguably the most vital organs in the gut for nutrient absorption. Notably, intraspecific variation in the small intestine can be associated with life history traits. In humans, females demonstrated consistently and significantly longer small intestines than males ($t$-value$_{15}$ = 2.245, $p = 0.0403$). This finding supports the female canalization hypothesis, specifically, increased female investment in the digestion and absorption of lipids.

## INTRODUCTION

### Scientific importance of variation

Morphology varies among individuals within species, and this variation is key to adaptation and evolution (*Darwin, 1859*). When variation among individuals results from genetic differences, that variation is key to natural selection. But even when variation is not genetic, or results from a mix of genes and plasticity, it can have important ecological consequences. For example, phenotypic variation allows species to collectively take advantage of more resources and conditions than can any individual within the species (*Bolnick et al., 2007*). In humans and domesticated animals, such variation has the added importance of playing a role in medical treatment and intervention (*Raikos & Smith, 2015*). Yet, while variation in many traits, including the shape of finch beaks (*Abzhanov, 2004*) and the shape of human noses (*Maddux et al., 2017*; *Zaidi et al., 2017*), has been described in great detail, some traits remain poorly studied, particularly the subset of internal soft tissue traits that are neither outwardly visible nor preserved in fossils. The gastrointestinal (GI) tract represents an organ system that directly impacts a species' ability to occupy a given niche space and also to adapt to changing conditions through targeted digestion of different nutrients (Table 1). The morphology of the GI tract is particularly important given recent revelations about the contributions of gut microbial communities to host health. But the extent to which different components of the GI tract vary among individuals is still poorly documented, impeding our understanding of why this variation exists and how it impacts microbial activity.

The lack of GI data is particularly apparent in humans. Early research on humans emphasized comparative morphology, including research focused on organs (*Treves, 1885*); but with time the study of morphology waned in popularity (for many reasons, including the advancement of molecular techniques), with the handful of subsequent studies of gastrointestinal morphology focusing almost exclusively on intestinal length (*Dreike, 1894*; *Bloch, 1904*; *Bryant, 1924*; *Blankenhorn, Hirsch & Ahrens, 1955*; *Underhill, 1955*). Indeed, the latter studies calculated the average length of the small intestine and colon but did not record inter-individual variation.

Dissection is the primary vehicle for students to form an intimate familiarity with the internal anatomy of humans and other animals (*Gunderman & Wilson, 2005*), but it can also present an opportunity to appreciate and quantify morphological variation. Here we hypothesize that the measurements taken by students, particularly medical students, can play a key role in increasing our understanding of variation in human GI tracts. We argue that such measurements have at least three potential values. (1) They provide a context in which medical students can pay attention to variation and its importance for medical treatment. (2) They can inform our understanding of human evolution and variation. Finally, (3) they can allow tests of general theory regarding the evolutionary and ecological tradeoffs among organ systems. We consider each framework below and in Table 2. If we are right, many thousands of thoughtfully donated human bodies could be put to even more use each year in the United States (and more globally) than is currently the case.

McKenney et al. (2023), *PeerJ*, DOI 10.7717/peerj.15148

**Table 1 Digestive processes and feeding strategies associated with different sites along the gastrointestinal tract.**

| Digestive site | Functional role | Macronutrient(s) digested | Nutrient(s) absorbed | Associated feeding strategy |
|---|---|---|---|---|
| Mouth | Mastication | Starches | – | Frugivore, omnivore, herbivore |
| Stomach | Acid catabolism | Proteins | – | Carnivore |
| Liver | Enzyme production, metabolism and detoxification, glycogen and triglyceride storage | Carbohydrates, lipids | – | Frugivore, omnivore, herbivore |
| Small intestine | Bile salt catabolism, absorption | Protein, lipids, carbohydrates | Carbohydrates, proteins, lipids | Carnivore, omnivore |
| Cecum, appendix | Microbial storage and fermentation | Fiber | Vitamins, short chain fatty acids, water | Folivore, omnivore |
| Colon | Microbial fermentation, water absorption | Fiber | Vitamins, short chain fatty acids, water | Folivore, omnivore |

McKenney et al. (2023), *PeerJ*, DOI 10.7717/peerj.15148

**Table 2  Different frameworks and predictions for anatomical variation in the current study.**

| Framework | Assumption / hypothesis | Prediction |
|---|---|---|
| Medical training | A lack of variational knowledge among medical students can lead to radiological misdiagnoses of variants as pathologies (*Raikos & Smith, 2015*). | Having students take measurements during dissections can reveal underappreciated variation in human bodies and the bodies of other frequently dissected animals. |
| Human evolution and variation | We hypothesize that the relative lack of data on modern human GI tracts hides heretofore unrecognized complexities regarding the evolution and ecology of human GI tracts. | We expect the present study to reveal greater human variation than previously reported in the literature, by virtue of adding more samples compared to previous studies, with consequent effects on our understanding of the evolution of human gut size and morphology. |
| General theory and organ size | Deviations from the mean are more likely to be costly in traits that are vital to survival and thus under stronger selective pressure. | We predict that the small intestine and colon exhibit less interindividual variation than other parts of the GI tract because they perform the most vital functions (*i.e.*, nutrient absorption; see Table 1). |
| Female canalization | Females might leverage developmental stability to reduce morphological variation and protect reproductive investment (*Ross, Baker & Falsetti, 2003*; *Willmore, Young & Richtsmeier, 2007*; *Moore, 2013*). | We predict that female humans will exhibit consistent differences in GI measurements and/or variation compared to males. |
| Concerted evolution hypothesis | Any single change or acquisition will affect all regions of the gut due to the developmental and physiological connections between them, resulting in coordinated variation among sites (*Finlay & Darlington, 1995*). | Within species, we would expect to detect strong correlations between different organs to coordinate the metabolism and absorption of nutrients. |
| Mosaic evolution hypothesis | Selection can drive the separate evolution of specific [gut] sites without resulting in system-wide changes (*Barton & Harvey, 2000*). | Because each site along the GI tract is specialized to target the digestion of specific macronutrients (see Table 1), we might predict that GI morphology would vary by organ among species, reflecting niche (*Violle et al., 2012*), defined here as feeding strategy. We might also expect to observe sex-specific differences in variation of specific gut sites related to differences in dietary requirements for reproductive purposes (*e.g.*, canalization in females related to the dietary investments for gestation and lactation). |

### Medical training and variation

One of us (Dunn) has argued that the focus on the "normal" or typical representative case and the mean is problematic in general in education (*Dunn et al., 2016*); but this approach is especially problematic for medical students. Lacking information on or even the awareness of normal human variability may lead to misdiagnosis and malpractice (*Willan & Humpherson, 1999*; *Granger, 2004*; *Nzenwa, Iqbal & Bazira, 2023*). We hypothesize that having students take measurements during dissections can reveal underappreciated variation in humans and other frequently dissected animals. We further hypothesize, but do not test here, that by paying attention to variation, medical students are more likely to be aware of such variation in the bodies of their own patients and the implications of such natural variation as it relates to treatment.

### Human evolution and variation

Roughly 1.9 million years ago (*Domínguez-Rodrigo, 2002*), humans have been variously hypothesized to have begun to use tools to eat shellfish (*Broadhurst, Cunnane & Crawford, 1998*) and honey (*Crittenden et al., 2011*); hot springs to cook foods (*Sistiaga et al., 2019*); and fermentation (*Amato et al., 2021*) and cooking (*Wrangham, 2009*) to transform foods. The expensive tissue hypothesis (*Aiello & Wheeler, 1995*) posits that this food transition fueled the evolution of a bigger brain in ancient humans *via* more available calories and more easily digested foods. Less chewing and less robust jaws freed space for a bigger brain; and long, large intestines were less necessary because more processed foods required less microbe-mediated digestion in the large intestine (*Aiello & Wheeler, 1995*; *Zink & Lieberman, 2016*). Yet, while our understanding of dental and jaw bone changes across human evolution is robust (*Organ et al., 2011*), our understanding of GI changes is not. This is in part because GI tissue is not preserved in ancient human fossils, and also because existing studies of the expensive tissue hypothesis focus on the shape and size of the rib cage (*Aiello & Wheeler, 1995*). We expect the present study to reveal greater human variation than previously reported in the literature, by virtue of adding more samples compared to previous studies, with consequent effects on our understanding of the evolution of human gut size and morphology.

Some GI structures may vary more than others, and this variation is potentially predictable. Variation might also differ predictably among species based on body size, evolutionary relatedness, feeding strategy, population size, or other species-level attributes. That is to say, variation might be a trait that is, itself, conserved in certain lineages as a result of multiple non-exclusive causes including genetic differences, developmental differences, or phenotypic plasticity. Some organisms respond plastically to environmental changes, in ways that buffer the change (*Willmore, Young & Richtsmeier, 2007*). This phenotypic response, known as canalization, increases short-term fitness benefits and maximizes reproduction (*Careau, Buttemer & Buchanan, 2014*). Canalization can also differ among sexes: it has been hypothesized that female humans are better able to buffer environmental challenges (*Bogin, 1999*; *Stinson, 2012*). We might then expect to observe consistent differences in the size or variation of specific GI measurements in females compared to males.

Here we propose a novel approach to simultaneously expand our understanding of variation among individuals within species and engage students in a better understanding of variation. First, we developed a citizen science-based approach wherein undergraduate students enrolled in biological courses at North Carolina State University measured key features of GI morphology as a part of a comparative dissection lab. The goal of this first component is to understand differences among species, and whether students can effectively record data on morphological variation as a part of traditional laboratory university coursework. Second, two experts (Hale and McKenney) measured variation in human cadavers used in gross anatomy courses at Duke University School of Medicine. As part of in-class activities, graduate and medical students also measured the same cadavers. If variation exists in human cadavers and other species, and if students can be a part of the process of recording such data, then our understanding of morphological variation could expand rapidly with as many as 10–12 million animal specimens dissected each year in classrooms (*Oakley, 2012*; *Dunn et al., 2016*). Here we (a) measured among species differences to determine whether they exist after body size is accounted for; (b) measured within-species variation to test whether different organs vary differently; and (c) measured among individual variation to determine if sex is explanatory.

## MATERIALS & METHODS

The raw data collected for this study and R code used to analyze the data are available in the Dryad repository (https://datadryad.org/stash/share/zqWgXsGeOTJ-zw5E_AaDLmmq1n3hGz9oKFx77QhiDjc).

### Data collection

Measurements were collected from formalin-preserved human cadavers and animal dissections to quantitatively compare GI variation within and across species. The cadaver data for 45 individuals (21 females and 24 males) were collected by the authors (EAM and ARH) at the Duke University School of Medicine gross anatomy laboratory. These cadavers were part of the medical school anatomy laboratory courses and the Duke Anatomical Gifts Program, for which individual donors consent to educational use, so IRB approval and consent were not needed. The age, sex, and cause of death for each individual are the only information provided. The full list of cadaver measurements can be found in Text S1. Measurements to the nearest 0.01 cm were recorded with the following tools: sliding calipers, spreading calipers, ruler tape, rulers, and string (see supplementary information for details).

Animal dissection data from 30 dissection specimens (10 rats [*Rattus norvegicus*], 10 pigs [*Sus scrofa*], and 10 bullfrogs [*Lithobates catesbeianus*]; Carolina Biological Supply Company) were collected by undergraduate students enrolled in a comparative anatomy laboratory course at North Carolina State University. The dissection specimens used in this study were not bred for research and were purchased through a third party for educational labs; therefore, no IACUC review was required by NC State. This lab was specifically designed to teach quantitative comparisons of the digestive system across species and to illustrate the potential effects of natural selection due to feeding strategy. Students measured

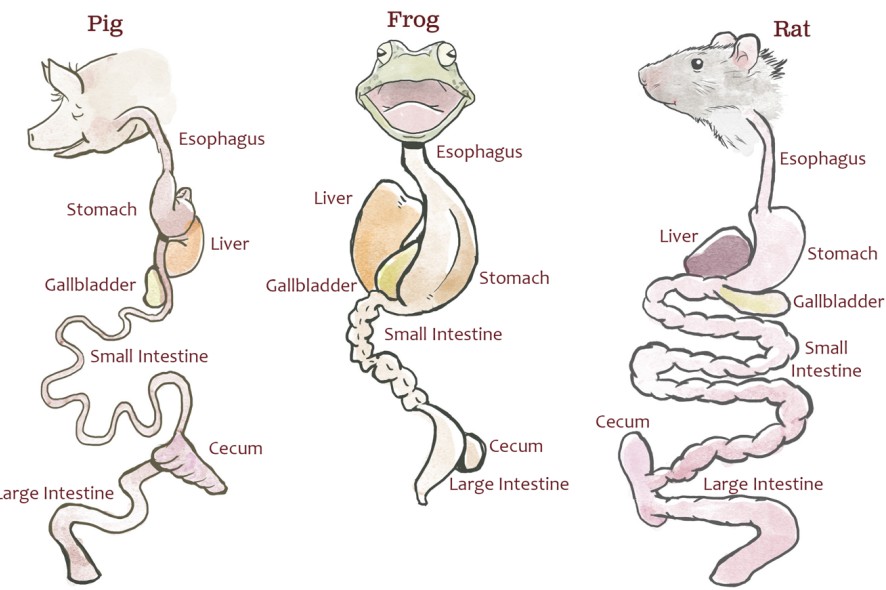

**Figure 1** **Representative gut diagrams for the fetal pigs (*Sus scrofa*), frogs (*Lithobates catesbeianus*), and rats (*Rattus norvegicus*).** Illustration by Chris Hedstrom.

each specimen's body length (from bregma to coccyx), esophageal length, stomach length (from fundus to pylorus), small intestine length, cecum length, and colon length (Fig. 1). The definitions provided to students for each measurement can be found in Text S2 and at http://studentsdiscover.org/lesson/a-diversity-of-guts/. Measurements were consolidated for instructional purposes.

Mean and standard deviations of the organ structures for the species examined in this study were also collected from published literature where available (see Tables S1 and S2). The aim was to investigate (1) how data were traditionally presented and (2) whether the variation observed in the study sample was comparable to published literature. ANOVA tests were implemented to identify significant differences in observed variation where information on the standard deviation of the samples was available.

## Comparative dissection GI analysis

Total intestinal length was calculated from the small intestine and colon lengths. Mean, standard deviation, minimum, and maximum values were calculated for each gut site. Coefficients of variation were calculated to measure the variability per site, within species and relative to their respective sample population means. Bartlett tests were performed to assess the homogeneity of variances among species.

## Cadaver GI analysis

For brevity, we reduced the cadaver variables to include only those also collected during animal dissections (see list above and Text S2), as well as those that exhibited the greatest variation (*i.e.,* liver volume, gallbladder max length, appendix length). We calculated the mean, standard deviation, minimum, and maximum values to provide ranges for each of

the measurements selected. Coefficients of variation were calculated to assess investment and variation in major digestive sites of interest. Sample sizes vary because for some individuals, the structure was either absent (*e.g.*, gallbladder, appendix) or unavailable for measurement. None of the human gut measurements collected were found to be correlated with cadaver height (tall humans did not have predictably longer guts than short humans); therefore, standardization by cadaver height was not performed for the following tests. All data were normalized to a mean of zero and a standard deviation of one to compare measurements across anatomical sites. We chose to employ Pearson product-moment correlations to test for significant relationships between each pair of variables. Linear regression was performed to measure the strength of relationships between variables. Analysis of variance (ANOVA) was performed on a logistic regression model to determine if significant sex differences were independent of the ranking of variables in the model. One-way ANOVA was also performed to test for significant variation between sexes for each variable. For those measurements that showed a significant relationship with sex, boxplots were used to visualize sex differences.

## RESULTS

### Student measurements increase our understanding of variation

Undergraduate students and experts recorded different gut lengths and more variation than has been previously reported in the published literature, across rat (*Rattus norvegicus*), frog (*Lithobates catesbeianus*), and fetal pig (*Sus scrofa*) specimens (Figs. 2 and 3; Tables S1 and S2). As predicted, we find that the magnitude of variation (*i.e.*, standard deviation) increased with sample size, in all species except for fetal pigs (*Sus scrofa*; Fig. 3).

Across 45 humans, we detected similar mean gut measures (Fig. 2D), consistent variation in small intestine length ($F_{4,32} = 0.2629$, $p = 0.2009$) but greater variation in colon length ($F_{9,22} = 0.11664$, $p = 0.002219$) compared to Hirsch (*Blankenhorn, Hirsch & Ahrens, 1955*) in his study of 10 individuals. Variation among individuals does not appear to have been reported or even discussed in detail in any of the other early studies of the morphology of the human GI tract. It is worth noting here that we found such variation even though we measured just 45 human cadavers.

The variability of small intestine, colon, and total intestinal length differed significantly between frogs and all other species (Tables 3 and 4). Bartlett tests demonstrated that the patterns of variation were different for each species. However, human measurements drove this heterogeneity: $F$-tests confirmed that only humans differed significantly from all other species for small intestine $F_{3,59} = 130.3$, $p < 2e{-}16$), colon ($F_{3,49} = 128.8$, $p < 2e{-}16$), and total intestinal length ($F_{3,49} = 37.56$ $p = 9.46e{-}13$).

We did not detect significant correlations in size among organs within rats, pigs, or frogs (Table S3). However, Pearson correlations and paired t-tests show that liver volume was significantly correlated with the length of appendix ($t$-value $= 2.5278$, $df = 28$, $p = 0.0174$, corr $= 0.4311$) and colon ($t$-value $= 2.0991$, $df = 19$, $p = 0.0494$, corr $= 0.4339$) in humans (Table 5, Table S4). This might be expected if the same factors that favor large microbial communities also favor large livers. We also detected a significant correlation between

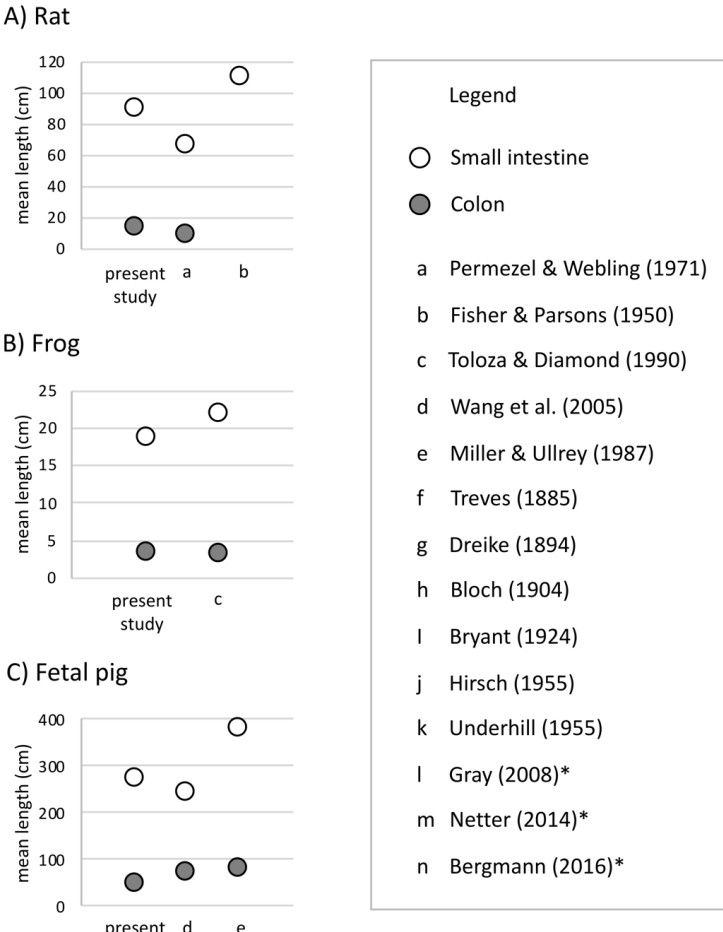

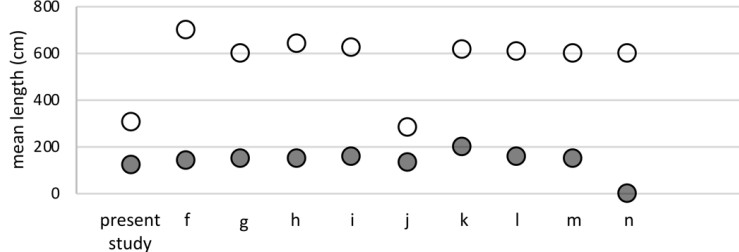

**Figure 2** **Mean length (in cm) of small intestine, cecum, and colon in (A) rats (*Rattus norvegicus*), (B) frogs (*Lithobates catesbeianus*), (C) fetal pigs (*Sus scrofa*), and (D) humans (*Homo sapiens*).** Empty shapes indicate small intestine measurements; filled shapes indicate colon measurements. Previously published data reported from *Permezel & Webling (1971)*, *Fisher & Parsons (1950)*, *Toloza & Diamond (1990)*, *Wang et al. (2005)*, *Miller & Ullrey (1987)*, *Treves (1885)*, *Dreike (1894)*, *Bloch (1904)*, *Bryant (1924)*, *Blankenhorn, Hirsch & Ahrens (1955)*, *Underhill (1955)*, *Standring, Borley & Gray (2008)*, *Netter (2014)*, and *Tubbs, Shoja & Loukas (2016)*. Asterisks indicate medical texts.

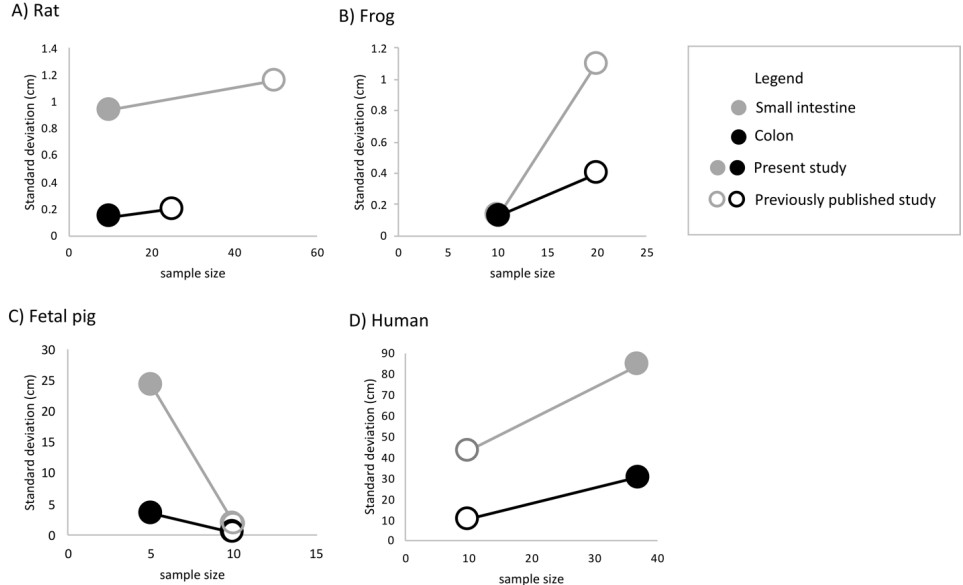

**Figure 3** **The amount of variation detected for (A) small intestine and (B) colon length increases with sample size for all species except fetal pig.** Empty shapes denote previously published studies (*Permezel & Webling, 1971*; *Fisher & Parsons, 1950*; *Toloza & Diamond, 1990*; *Wang et al., 2005*; *Blankenhorn, Hirsch & Ahrens, 1955*). Filled shaped denote the present study.

**Table 3** **ANOVA results comparing variation in small intestine length, colon length, and total intestinal length between carnivorous (*i.e.*, frog) versus omnivorous (*i.e.*, rat, pig, and human) specimens.** Cecum length was excluded because the single carnivore has no cecum. Significant *p*-values are in bold ($\alpha = 0.05$).

| Variable | df | SS | F-statistic | *p*-value |
|---|---|---|---|---|
| Small intestine length | 1, 61 | 5.76e5 | 15.01 | **2.6e−4** |
| Colon length | 1, 51 | 4.24e5 | 10.23 | **2.4e−3** |
| Total intestinal length | 1, 51 | 3.33e6 | 8.41 | **5.5e−3** |

small intestine and colon length (*t-value* $= 2.1699$, *df* $= 17$, $p = 0.0445$, Table 3, Table S4). Most other structures did not exhibit such correlation: for example, the size of the cecum could not be predicted based on the size of the small intestine.

Our results suggest that organs behave independently of one another, both within and across species. Bartlett tests confirmed that stomach length exhibited more homogenous variance among species compared to other organs (*Bartlett's k-squared* $= 3.4495$, *df* $= 2$, $p = 0.1782$, Table 6). In rats (*Rattus norvegicus*), which are generally considered to be omnivorous, the length of the cecum is more variable than that of the colon (Table 4). Liver volume exhibited the greatest variability in humans, varying by as much as 54% among individuals (Table 7). The next most variable structures were those that house microbial communities (*e.g.*, cecum and appendix).

**Table 4** Variation across three commonly dissected species in select GI measurements in centimeters.

**Frog (_Lithobates catesbeianus_)**

| Measurement | n | Mean (cm) | Min | Max | Std Dev | $C_v$ |
|---|---|---|---|---|---|---|
| Body Length | 10 | 9.875 | 8.7 | 11.5 | 0.882 | 8.93% |
| Length of Esophagus | 10 | 2.545 | 1.75 | 3.7 | 0.078 | 27.67% |
| Length of Stomach | 10 | 4.50 | 3.3 | 6.8 | 0.129 | 21.57% |
| Length of Small Intestine | 10 | 19.080 | 16.2 | 22.7 | 0.133 | 9.40% |
| Length of Cecum | – | – | – | – | – | – |
| Length of Colon | 10 | 3.49 | 2.5 | 6.9 | 0.133 | 36.56% |

**Pig (_Sus scrofa_)**

| | | | | | | |
|---|---|---|---|---|---|---|
| Body Length | 10 | 32.58 | 25 | 36 | 3.46 | 11.18% |
| Length of Esophagus | 10 | 9.889 | 3 | 15.5 | 0.115 | 35.73% |
| Length of Stomach | 10 | 8.167 | 5 | 15.25 | 0.096 | 40.47% |
| Length of Small Intestine | 10 | 283.8 | 220 | 385.5 | 1.9 | 16.31% |
| Length of Cecum | 10 | 4.033 | 2.8 | 6 | 0.025 | 25.07% |
| Length of Colon | 10 | 50.19 | 29.5 | 69.5 | 0.383 | 23.56% |

**Rat (_Rattus norvegicus_)**

| | | | | | | |
|---|---|---|---|---|---|---|
| Body Length | 10 | 16.180 | 13.1 | 19 | 1.905 | 11.78% |
| Length of Esophagus | 10 | 6.685 | 3 | 10.7 | 0.192 | 35.38% |
| Length of Stomach | 10 | 7.12 | 3.5 | 11.5 | 0.182 | 35.51% |
| Length of Small Intestine | 10 | 91.260 | 81 | 102.9 | 0.942 | 8.33% |
| Length of Cecum | 10 | 9.215 | 5 | 13 | 0.202 | 29.11% |
| Length of Colon | 10 | 14.41 | 12.5 | 16.9 | 0.14 | 9.60% |

**Table 5** Pearson product-moment correlation for cadaver dataset for each of the measurements examined. Correlations are in lower diagonal and p-values are reported in upper diagonal. Significant $p$-values are in bold ($\alpha = 0.05$). Test statistics and degrees of freedom are reported in Table S5.

| | Liver volume | Length of Gallbladder (maximum) | Length of Cecum | Length of Small Intestine | Length of Appendix | Length of Colon |
|---|---|---|---|---|---|---|
| Liver volume | 0 | 0.7242 | 0.0767 | 0.2978 | **0.0174** | **0.0494** |
| Length of Gallbladder (maximum) | −0.0685 | 0 | 0.4234 | 0.4826 | 0.3400 | 0.7613 |
| Length of Cecum | 0.2796 | −0.2527 | 0 | 0.1646 | 0.3502 | 0.2804 |
| Length of Small Intestine | 0.1899 | −0.1505 | 0.2477 | 0 | 0.0755 | **0.0445** |
| Length of Appendix | 0.4311 | 0.1991 | 0.1679 | 0.3696 | 0 | 0.4399 |
| Length of Colon | 0.4339 | −0.0796 | 0.2350 | 0.4657 | 0.1830 | 0 |

## Evidence for female buffering

Overall, logistic regression suggests that the variation among individuals was not strongly correlated with the sex of the individuals (Table S5; $F_{3,17} = 1.228$; $p = 0.3303$). However, the logistic regression model for small intestine length with sex as a covariate showed a significant difference between males and females ($F_{3,15} = 88.31$, $p = 9.291e{-}10$). Specifically, we found that females have consistently and significantly longer small intestines compared to males (Fig. 4; $t$-value $15 = 2.245$, $p = 0.0403$).

**Table 6  Bartlett tests for homogeneity of variances among frog (*Lithobates catesbeianus*), pig (*Sus scrofa*), and rat (*Rattus norvegicus*) specimens.** Significant *p*-values are in bold ($\alpha = 0.05$).

| Variable | Bartlett's k-squared | df | *p*-value |
|---|---|---|---|
| Body Length | 13.504 | 2 | **0.001169** |
| Length of Esophagus | 6.7381 | 2 | **0.03442** |
| Length of Stomach | 3.4495 | 2 | 0.1782 |
| Length of Small Intestine | 35.11 | 2 | **2.376e−8** |
| Length of Cecum | 24.15 | 2 | **8.911e−7** |
| Length of Colon | 12.921 | 2 | **0.001564** |

**Table 7  Summary statistics for the GI organs across the human cadaver sample.** All measurements are provided in cm or cm$^3$.

| Measurements | n | Mean | Min | Max | Std Dev | $C_v$ |
|---|---|---|---|---|---|---|
| Length of Body | 45 | 168.2 | 149.0 | 184.0 | 10.4 | 6.20% |
| Liver volume[*] (cm$^3$) | 41 | 1174.83 | 377.38 | 2872.53 | 641.39 | 54.59% |
| Length of Gallbladder (maximum) | 30 | 8.85 | 5.50 | 12.50 | 1.65 | 18.62% |
| Length of Small intestine | 33 | 419.05 | 193.5 | 592 | 84.33 | 20.12% |
| Length of Duodenum | 33 | 27.20 | 17.1 | 46.3 | 7.37 | 27.11% |
| Length of Jejuno-ileum | 43 | 391.56 | 193.50 | 592.00 | 78.71 | 20.10% |
| Length of Cecum | 44 | 12.68 | 6.5 | 25.80 | 4.22 | 33.26% |
| Length of Appendix | 33 | 6.91 | 1.40 | 12.7 | 2.48 | 35.95% |
| Length of Colon | 23 | 134.73 | 80.9 | 199 | 30.75 | 22.83% |
| Total intestinal length | 23 | 486.04 | 80.9 | 758.5 | 193.06 | 39.72% |
| **Ratios** | | | | | | |
| Liver volume: small intestine | 33 | 3.10 | 0.95 | 6.07 | – | – |
| Duodenum: jejuno-ileum | 33 | 0.07 | 0.03 | 0.11 | – | – |
| Duodenum length:total small intestine | 33 | 0.06 | 0.03 | 0.10 | – | – |
| Total small intestine: total colon | 19 | 3.20 | 2.18 | 4.98 | – | – |
| Appendix length: total colon | 20 | 0.06 | 0.01 | 0.10 | – | – |
| Cecum length: total colon | 23 | 0.10 | 0.06 | 0.21 | – | – |
| Total small intestine: cadaver length | 32 | 2.50 | 1.25 | 3.86 | – | – |
| Appendix length: cadaver length | 32 | 0.04 | 0.01 | 0.08 | – | – |
| Cecum length: cadaver length | 43 | 0.07 | 0.04 | 0.13 | – | – |
| Total colon length: cadaver length | 22 | 0.80 | 0.44 | 1.12 | – | – |
| Total intestinal length:cadaver length | 20 | 3.08 | 0.44 | 4.80 | – | – |

**Notes.**
[*]Calculated as $V = pi * r^2 (h/3)$ where r = $(\frac{1}{2})$maximum length of liver and h = height of liver at the hilum.

## DISCUSSION

We measured the digestive systems in humans, rats, pigs, and bullfrogs to investigate morphological variation within and across species. Because variation is the raw material upon which natural selection acts, quantifying variation across traits and species lends insight to evolutionary processes and adaptive flexibility (*i.e.,* through phenotypic plasticity). For lab animals (such as Norway rats) such variation may influence experimental

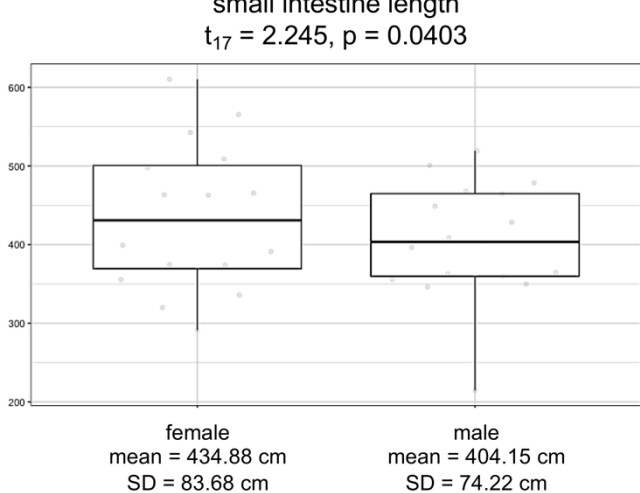

**small intestine length**
$t_{17} = 2.245$, p = 0.0403

female
mean = 434.88 cm
SD = 83.68 cm

male
mean = 404.15 cm
SD = 74.22 cm

**Figure 4** **Boxplots by sex for small intestine length, in centimeters.** Model significance is based on $\alpha$ =0.05.

responses. For domestic animals (such as pigs), it may influence growth rate and health. For humans, variation may further drive or contribute to individual responses to diets, microbiome compositions, or medical conditions, ultimately influencing diagnosis. Given the great genetic and ancient dietary diversity of humans, as well as the potential for variation due to plasticity in some human traits, and the medical consequences that variation is known to have, it is important to learn about variation, especially in humans. For example, knowledge of variation in the insertion of the male gonadal vein can assist in distinguishing conditions such as pelvic congestive syndrome and infertility from normal variation (*Gupta, 2015*). Yet discussions of variability have been dropped from many medical texts over the years, and anatomy classes generally do not emphasize variation, much less its importance in contributing to health or disease states (*Nzenwa, Iqbal & Bazira, 2023*). (For example, an individual with a large appendix might be expected to host a larger reservoir of beneficial microbes, and to recover more quickly and completely from a diarrheal illness.)

We are beginning to understand the evolution of some genetic differences that influence digestion in humans. For example, early hominids evolved versions of alcohol dehydrogenase that increased their ability to metabolize alcohol some forty-fold (*Carrigan et al., 2015*). Similarly, early hominids also evolved novel lactic acid receptors, potentially in association with increased ingestion of lactic-fermented foods (*Amato et al., 2021*; *Davenport et al., 2017*). We also know that some later human populations evolved lactase persistence. Yet, we still know very little about larger structural changes in the guts of humans–and the inferences that have been made are based on the size of the rib cage of different human species (*Ben-Dor, Gopher & Barkai, 2016*; *Schmid et al., 2013*; *Williams et al., 2017*). In addition to assuming that rib cage size predicts gut morphology, such studies

assume a mean size of the rib cage for particular species of *Homo* without regard to native and climatic variation.

Here we present a novel approach that engages students in expanding our understanding of variation among individuals. We demonstrate that students can measure previously unreported variation, and that those data can inform medical diagnosis and treatment as well as our broader understanding of evolution.

## Medical training and variation

We first assessed the scientific value of student measurements collected during educational labs. As we predicted, the students with whom we worked uncovered previously unknown variation based on the study of a relatively small number of dissections. The Association of American Medical Colleges estimates that first-year medical school enrollment will reach 21,304 in 2019–2020. If we conservatively assume that 6 students share a single human cadaver, then more than 3,500 human cadavers are dissected each year in the US alone, and tens (if not hundreds) of thousands since cadaver dissection became a common feature of medical training (not to mention cadavers dissected by dentists, undergraduates, *etc.*). The actual variation among human bodies seen (but not documented) by students each year is therefore undoubtedly much greater than what we have documented here; and notably, *via* the approach we used here, that variation is documentable.

## Human evolution and variation

We next compared our results to previous studies. Even based on a small sample of modern humans, we are able to show much more variation in gut morphology than heretofore noticed. Where previous studies have compared the relative size of large intestines to small intestines in humans and extant ape species (as a measure of the degree of specialization on gut-fermented foods), those studies have included data from just six humans, described as "urban males", all from a single population (*Ragir, Rosenberg & Tierno, 2000*). Those six human males from a single, urban population were then compared to data from a single chimpanzee (*Chivers & Hladik, 1980*). Given the extraordinary diversity of modern humans, we suspect that the diversity we have documented is just a small portion of what exists globally. Some of this variation is likely to be due to plasticity, but genetic differences also likely play a role.

## General theory and organ size

We next compared variation across species to test whether different organs vary in different ways. Different animal species have evolved a variety of gastrointestinal morphologies to facilitate different feeding strategies (*Stevens & Hume, 1998*) through differential investment in tissues tailored to digest and absorb specific nutrients (see Table 1).As predicted, omnivorous pigs (*Sus scrofa*) and rats (*Rattus norvegicus*) exhibit greater variation in the length of the small intestine and colon compared to carnivorous frogs (*Lithobates catesbeianus*), which are nutritional specialists by comparison (Table 4). The generality of conclusions we can draw from this comparison is limited: we considered just three omnivores and one carnivore, the latter of which also happened to be the only non-mammal in the study. Yet, our results demonstrate that further study is warranted.

Humans exhibit increased variation in intestinal length compared to other species, perhaps because humans do not consume a standardized captive diet. This explanation is supported by a previous study, in which Saric and colleagues (*Saric et al., 2008*) characterized fecal metabolites in humans, rats, and mice, and found that humans exhibit the greatest intraspecific variation. These results, combined with differences in captive diet, suggest that dietary variation may beget GI variation. Future studies could incorporate stable isotope analysis to correlate gut length variation with diet (*Kaehler & Pakhomov, 2001*; *Kiszka, Lesage & Ridoux, 2014*; *Kishe-Machumu et al., 2017*). Differences across species may also be driven by life stage: Treves (*Treves, 1885*) and Zabielski (*Zabielski, Godlewski & Guilloteau, 2008*) both previously demonstrated differential and site-specific intestinal development in neonates compared to adults.

Interestingly, within humans, the total intestinal length exhibits less standardized variation (−0.5 to 1.5) than did either the small intestine or the colon, suggesting that there may be a tradeoff in investment between the two tissues (*i.e.,* when the colon is longer, the small intestine is shorter, and vice versa). Pearson's coefficient (Table 5) confirms a correlation between small intestinal and colon length (*p = 0.0445*) and may indicate concerted physiological adaptations (*Finlay & Darlington, 1995*) for greater investment either in protein absorption in the small intestine or in the microbial fermentation of dietary fiber in the colon (*Stevens & Hume, 1998*). Certainly, the correlations between liver volume and length of the appendix and colon warrant further study to test for a relationship between human and microbial metabolism, specifically if the variation observed in the cecum and appendix is mediated by microbial colonization during development.

While the stomach was the most variable organ in pigs (*Sus scrofa*) and rats (*Rattus norvegicus*), it exhibited similar variability across all species. This finding is of interest because the vertebrate stomach can take many forms (*Wilson & Castro, 2010*). Our results suggest that, while the stomach varies within the species presented here, that variability does not differ significantly among species. Notably, the variation within humans compared between this study and Hirsch (*Blankenhorn, Hirsch & Ahrens, 1955*) was consistent for the small intestine, but not the colon; these results are consistent not only with the different functions of these two intestinal tract sites, but also with their developmental control.

Most catarrhine primate digestive systems are characterized by a simple stomach, a C-shaped duodenum, and a more globular and reduced cecum with increased emphasis on microbial fermentation in the colon (*Stevens & Hume, 1998*; *Lambert, 2002*). This is due, in large part, to a primarily herbivorous diet, which is quickly passed through the stomach and duodenum before slowing in the ileum and cecum (*Ragir, Rosenberg & Tierno, 2000*). By comparison, humans consume a greater diversity of food items comprising less fiber and more protein and fat; yet the only substantive morphological difference is a longer small intestine in humans (*Carmody & Wrangham, 2009*; *Watkins et al., 2010*). Chimpanzees do occasionally consume animal protein, but exhibit little significant difference in food retention and processing times compared to other catarrhine primates (*Milton & Demment, 1989*; *Lambert, 2002*; *Carmody & Wrangham, 2009*). In this study, we found that the duodenum—the proximal segment of the small intestine, which is involved in digestion—varies less than the jejuno-ileum—the distal segment, involved in

absorption. When combined with what is known about other primates, this could suggest that the duodenum is under greater developmental control because it functions more in protein digestion, which has greater importance for humans. Humans also consume higher-quality diets with cooking substituted for some chemical digestion (*Carmody & Wrangham, 2009*). Further, the increased variability of the human colon, similar to other primates (*Lambert, 2002*), may indicate increased plasticity in response to differential consumption of high-fiber foods.

Individuals' proclivity for lipid metabolism may remain constant, or it may be regulated by the liver and gallbladder, which exhibit greater variation, while jejunoileal length may be determined by the nutritional composition (*i.e.,* fat content) of the diet. The greater variance documented in human liver volume may result from age differences and body proportions (*Wynne et al., 1989*), or it may be an artifact of the multiple measurements that contributed to the geometric proxy for volume.

### Evidence for female buffering in humans

Finally, we investigated patterns in human morphological variation. We expected to find sex-specific differences in investment of digestive tissues, which would increase absorption and assist the immune system in females as predicted by proponents of female canalization (*Stinson, 2012*; *Careau, Buttemer & Buchanan, 2014*). We found that females consistently exhibited greater small intestinal length than did males, suggestive of increased investment in fat absorption in females. Additionally, the low variation we detected in small intestinal length suggests that these sex-specific differences are highly constrained. As a means of long-term energy storage and a major component of breast milk, fat is crucial for supporting both reproduction and lactation in females. By contrast, males exhibited greater colon and cecum lengths compared to females, indicating both that total intestinal length is driven by small intestinal length, and also that males may compensate for decreased fat absorption with increased dependence on microbial fermentation. Microbial fermentation is especially important to process "low quality" diets that contain more fiber and less fat or other easily digestible nutrients (*Stevens & Hume, 1998*). Males also tend to have larger livers than females, possibly to metabolize microbial fermentation products, though the difference in size could also be explained by general differences in body and abdominal cavity size canalization. Regardless, to our knowledge, these results comprise the first evidence of female buffering achieved through gastrointestinal morphology.

## CONCLUSIONS

Together, our results support both the mosaic and concerted hypotheses of evolution: we find that animal species differentially invest in specific digestive tissues; and, within species, we detected significant correlation between sites. Notably, humans exhibit increased variation in intestinal length compared to other species. One explanation for this increased variation may be that humans do not consume a standardized captive diet, as is the case for the other species dissected in this study. However, differences across species may also be driven by life stage. Future studies and additional data are needed to further investigate the factors that contribute to both inter- and intraspecific variation.

Comparative analyses can also inform our understanding of evolutionary history and the role of particular traits. Historically, morphological differences have played an important role in the classification and inference of phylogenetic relationships. However, this paper clearly demonstrates the need for additional morphological data to better interpret the developmental and functional aspects of variation.

## ACKNOWLEDGEMENTS

We thank Leonora Shell for coordinating the art, formatting the lesson plan, and offering suggestions to improve the dissection measurement protocol. We also thank Leonora Shell and Lauren Nichols for photographing the measurement process.

### Funding

This work was funded by NSF grant #1319293. The funders had no role in study design, data collection and analysis, decision to publish, or preparation of the manuscript.

### Grant Disclosures

The following grant information was disclosed by the authors:
NSF: 1319293.

### Competing Interests

The authors declare there are no competing interests.

### Author Contributions

- Erin A. McKenney conceived and designed the experiments, performed the experiments, analyzed the data, prepared figures and/or tables, authored or reviewed drafts of the article, and approved the final draft.
- Amanda R. Hale conceived and designed the experiments, performed the experiments, analyzed the data, prepared figures and/or tables, authored or reviewed drafts of the article, and approved the final draft.
- Janiaya Anderson performed the experiments, prepared figures and/or tables, authored or reviewed drafts of the article, and approved the final draft.
- Roxanne Larsen conceived and designed the experiments, performed the experiments, authored or reviewed drafts of the article, and approved the final draft.
- Colleen Grant conceived and designed the experiments, performed the experiments, prepared figures and/or tables, and approved the final draft.
- Robert R. Dunn conceived and designed the experiments, authored or reviewed drafts of the article, and approved the final draft.

### Data Availability

The raw data and R code used in this study are available at Dryad McKenney, Erin et al. (2023), Data for: Hidden diversity: Comparative functional morphology of humans and other species, Dryad, Dataset, https://doi.org/10.5061/dryad.4qrfj6qdj.

## Supplemental Information

Supplemental information for this article can be found online at http://dx.doi.org/10.7717/peerj.15148#supplemental-information.

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
