# Peer review of "Hidden diversity: comparative functional morphology of humans and other species"

_PeerJ, doi:10.7717/peerj.15148_

## Round 0.1 · original submission · Minor Revisions

Dear Authors,

Our expert reviewer panel appreciates the study mentioned in the manuscript. However, it has a few issues to be addressed. Therefore, I invite you to carry out minor revisions and to submit the manuscript as soon as possible.
Best of luck

·

Basic reporting

No comment

Experimental design

No comment

Validity of the findings

No comment

Additional comments

Being an Anatomist, I feel that the work is really interesting and the facts are really very much useful in clinical correlation between species and between sex.

·

Basic reporting

In the present investigation, the authors have shown how that morphological variation in the GI tract of humans and other species is associated with each organ’s role in food processing. They have done a basic, extensive, informative, and detailed about inter-species and within-species variations.

Experimental design

The experiment is well-designed and executed, and the manuscript is well-written. However, there are specific observations and suggestions from the side to improve the quality of the manuscript.

Validity of the findings

Data has been presented well with the help of graphs and tables.

Additional comments

1. Introduction: Extensively written and justifies the very purpose of the said experiment. But that could be made more concise, and much of the review part could be used in discussion or should be presented in tabular form.
2. Results: Drawings or original pictures of the organs in the study of various species with labeling will also help the readers in general about the gross morphological variations.
3. Abstract: If the authors could provide a graphical abstract of the manuscript, it would improve the overall understanding of the

·

Basic reporting

no comment

Experimental design

no comment

Validity of the findings

no comment

Additional comments

The manuscript addresses a fundamental but mostly ignored area of morphological variation and its clinical and evolutionary significance, and the authors deserve great credit for taking on this work.

---

## Round 0.2 · Minor Revisions

Dear Authors,

As the corrections made by you during previous revisions were checked by our expert reviewers and they really appreciated your efforts to match journal’s standards. However, one of our reviewers feels that still few more points to be addressed to improve the quality of the manuscript.

Therefore, as the Academic Editor for your article I wish you to invite for Minor Revisions. Please do the needful and resubmit asap.

Best of luck

·

Basic reporting

The revised manuscript has incorporated the suggestions to improve its clarity. Reader comprehension of the manuscript's message has also been enhanced by the addition of a graphical abstract.

Experimental design

It looks good, describes the various aspects of the experiment, and provides a clear message about the research question and methods to address the same.

Validity of the findings

The data recorded in the experiment has been statistically analyzed and well presented.

Additional comments

The frameworks, presumptions, hypotheses, and predictions examined in this study are listed in a new Table 2 by the authors. This has made the introduction part more precise, to the point, and easier to understand.
As suggested, the descriptive information that included summaries of pertinent literature was moved to the discussion to further streamline the introduction.

·

Basic reporting

The language of the manuscript is up to the standards.

The structure and presentation of the manuscript require some modifications.
Introduction
Most humbly it is brought to your kind notice that the introduction was found to be a little vague. It does introduce the topic however is not focussed with regards to the study being conducted. The objectives are not clear according to the results and discussion. The following suggestions may be followed for rewriting the introduction:
• first the topic is introduced with standard but brief chronological review up to recent times,
• Based on introduction, then the problem/ lacuna in research is recognized and hypothesis is put forward.
• Then based on hypothesis, objectives of the investigation are given.
Literature well referenced & relevant..

Figures are relevant, high quality, well labelled & described properly , however, kindly check the numbering sequence.

Raw data was supplied

Experimental design

The presented research is original primary research within Scope of the journal.

Research question were not well defined in the introduction or material and methods however are relevant & meaningful.

It is stated how the research fills an identified knowledge gap.

Rigorous investigation was performed to a high technical & ethical standard.

The material and method may kindly mention a statement about following the guidelines of ethical treatment to animals.
Kindly include a statement in the beginning of each protocol emphasising its relevance to the study.
Kindly include a brief statement about from where the animal and human cadavers were obtained.
Kindly mention if the cadavers were subjected to fixatives/ preservatives.

Validity of the findings

The research will have good impact in concerned field and it is a unique novel work.

Following observation regarding discussion are mentioned.
Most humbly it is brought to your kind notice that the discussion was found to be vague and inconclusive. The following suggestions may be followed for rewriting the discussion:
• First the topic is introduced by mentioning the hypothesis and objectives in brief with standard but brief chronological review up to recent times. For example “the presented study was conducted to investigate ………”. Followed by brief review “earlier research had showed …………”
• Results may kindly be discussed one by one leading to final conclusion
• The discussion of each finding may kindly start with mentioning the reason why the test was conducted or by stating the relevance of the test conducted for the presented study. If required a citation may be given.
• The discussion of each finding may kindly be clear cut and direct in comparison. Each finding discussion should end in a conclusive statement like- “the findings of the test clearly indicate that ……………” . At this point the novelty of findings may be mentioned against the cited research to show the impact of presented research.
• Discussion may end with remarks briefly mentioning the finding in logical sequence to draw a meaningful conclusion.

Reviewer 5 ·

Basic reporting

The article uses clear language that is simple to comprehend. The results are discussed and explained, and the hypotheses are well-presented. The authors have read a lot of literature on the subject and given references in order to explain their findings and give more information about it.

Experimental design

The experiment is well-designed, and the results are analyzed using suitable statistical tests.

Validity of the findings

No comments.

Additional comments

The authors have incorporated the suggestions by previous reviewers.

---

## Round 0.3 · Minor Revisions

Dear Dr. McKenney,

As the corrections made by you during previous revisions were checked by our expert reviewers and they really appreciated your efforts to match the journal’s standards. However, one of our reviewers feels that still a few more points to be addressed to improve the quality of the manuscript.

Therefore, as the Academic Editor for your article, I wish you to invite you to make Minor Revisions. Please do the needful and resubmit asap.

Good luck

·

Basic reporting

All corrections included

Experimental design

All corrections included

Validity of the findings

Conclusion may kindly be revised to include only concluding remarks without any references or citations.
All the comparisons with citations and reference to tables may kindly be removed.
All the comparisons to any previous study should be restricted to only discussion section.

Additional comments

The manuscript is almost complete with all corrections included, however only conclusions need some editing.

---

## Round 0.4 · accepted · Accept

Dear Dr. McKenney,

It is my pleasure to inform you that as per the recommendation of our expert reviewers, the manuscript "Hidden diversity: comparative functional morphology of humans and other species" has been Accepted for publication in PeerJ.

This is an editorial acceptance and you will be intimated for the list of further tasks before publication. So, I request you to be available for a few days to make the necessary things asap.

Regards and good luck with your future submissions.

·

Basic reporting

No comments

Experimental design

No comments

Validity of the findings

No comments

Additional comments

No comments